# Maternal Nutrient Excess Induces Stress Signaling and Decreases Mitochondrial Number in Term Fetal Baboon Skeletal Muscle

**DOI:** 10.3390/biology14070868

**Published:** 2025-07-17

**Authors:** Xu Yan, Carolina Tocantins, Mei-Jun Zhu, Susana P. Pereira, Min Du

**Affiliations:** 1Department of Animal Science, University of Wyoming, Laramie, WY 82071, USA; sean.yan@vu.edu.au (X.Y.); meijun.zhu@wsu.edu (M.-J.Z.); 2Institute for Health and Sport, Victoria University, Melbourne, VIC 8001, Australia; 3CNC-UC, Center for Neuroscience and Cell Biology, University of Coimbra, 3004-531 Coimbra, Portugal; ctsantos@cnc.uc.pt; 4CIBB, Center for Innovative Biomedicine and Biotechnology, University of Coimbra, 3004-531 Coimbra, Portugal; 5PDBEB—Doctoral Programme in Experimental Biology and Biomedicine, Institute for Interdisciplinary Research, University of Coimbra, 3004-517 Coimbra, Portugal; 6School of Food Science, Washington State University, Pullman, WA 99164, USA; 7UCIBIO, Applied Molecular Biosciences Unit, Department of Chemistry, Faculty of Science and Technology, NOVA University Lisbon, 2829-516 Caparica, Portugal; 8Associate Laboratory i4HB-Institute for Health and Bioeconomy, Faculty of Science and Technology, NOVA University Lisbon, 2829-516 Caparica, Portugal; 9Department of Animal Sciences, Washington State University, Pullman, WA 99164, USA

**Keywords:** maternal obesity, fetal programming, skeletal muscle metabolism, cellular bioenergetics, developmental programming, nutrient excess

## Abstract

Obesity during pregnancy can impact the development of the fetus and increase the risk of diseases throughout life, beginning from an early stage. We proposed that maternal obesity, caused by excess nutrient intake, affects the offspring’s skeletal muscle by causing inflammation and disrupting the normal function of mitochondria, essential structures for producing energy in cells. To investigate this, female baboons were fed either a normal diet or a nutrient-excess diet with increased fat and sugar content before and during pregnancy. Near the end of gestation, muscle samples from baboon fetuses from mothers on the nutrient-excess diet showed increased indicators of inflammation and decreased levels of key players for proper mitochondrial function. These changes suggest that maternal nutrient excess interferes with mitochondrial metabolism, which may raise the risk of metabolic diseases like diabetes early in life.

## 1. Introduction

Obesity is considered a global epidemic that has alarmingly reached middle- and low-income countries and affects men and women of all ages [1]. The prevalence of obesity in women of childbearing age is increasing [2]. According to the World Health Organization (WHO), 17.9% of women aged 18 and over were obese in 2022, and 44% were overweight [3]. In the United States (US), where obesity has reached staggering proportions, almost 42% of women aged 20 and over are obese, of whom around 40% are women between 20 and 39 years of age [4]. Moreover, the percentage of women in the US with pre-pregnancy obesity increased to 29% in 2019 (26.1% in 2016) [5]. This increasing obesity trend has also been observed among children [6]. The WHO estimated that 35 million children under 5 were overweight in 2024, and more than 390 million children and adolescents (5–19 years of age) were overweight in 2022, of which 160 million were obese [3].

Numerous epidemiological studies have shown that maternal weight and nutrition during pregnancy may have short- and life-long implications for the offspring’s health [7,8,9,10]. Increased maternal body mass index (BMI), either overweight (BMI ≥ 25 kg/m^2^) or obesity (BMI ≥ 30 kg/m^2^) was associated with adverse neonatal outcomes, including large for gestational age, macrosomia, and extreme pre-term birth [7]. Studies also associated maternal obesity (MO) with neonatal hyperinsulinemia [8] and hypoglycemia [8,9], independently of gestational diabetes mellitus. A multi-cohort meta-analysis estimated that up to 41.7% of the children born to mothers with pre-pregnancy overweight or obesity were themselves overweight or obese [8]. Maternal BMI was also strongly associated with early type 2 diabetes (T2D) and cardiovascular disease development in the offspring of the Helsinki Birth Cohort Study [10].

Fetal exposure to an adverse intrauterine environment can induce metabolic programming [11,12]. The skeletal muscle (SM) represents 40–50% of total body mass and has a paramount role in whole body postprandial insulin-stimulated glucose uptake and fatty acid utilization [13]. Given that muscle fibers do not increase in number after birth, poor fetal SM development may result in permanently reduced muscle mass and function, with long-term effects on body energy homeostasis, thus potentially representing an underlying cause for increased T2D predisposition. Lower muscle mass was associated with increased diabetes development [14,15], independent of body fat distribution, in young adults [14]. Considering the volume of epidemiological studies establishing associations regarding maternal nutrition–adverse intrauterine environment– and developmental disease, it has become clear that the molecular pathways underlying offspring disease must be tackled. Non-human primates (NHP) and other precocial species such as sheep [16] are excellent models to achieve the closest translation to human health and disease, given their similarities concerning genetics, physiology, immunology, the reproductive system, and developmental processes [17]. 

The effects of MO and nutrient-excess diet during pregnancy on fetal SM have started to be explored. Japanese macaques fetuses (130 days of gestation (d)) exposed to MO and western diet consumption during pregnancy (term 173 d) had reduced insulin-stimulated glucose uptake and Akt phosphorylation [18], which we also found in our study in the fetal SM of ewes at 135 d (term 148 d), similarly exposed to a nutrient-excess diet before and during pregnancy [19]. While neither study showed changes in phosphorylated insulin receptor substrate (IRS-1), Omar et al. reported an increased ratio of phosphorylated (inhibited) IRS-1 in the fetal SM of ewes mid-gestation (75 d), along with elevated active c-jun-NH_2_ terminal kinase (JNK) levels [20], which we also observed at late ewe gestation [19]. This kinase responds to inflammatory signals and inhibits insulin signaling [21]. Maternal obesity is characterized by a low-grade systemic and placental inflammation state [22]. Indeed, in our ewe model, toll-like receptor (TLR) 4 levels were increased, as well as the levels of phosphorylated TLR ligands activate IkB kinases (IKK) and nuclear factor-kB (NF-kB) p65 in the fetal SM [19], indicating an MO-induced inflammatory environment for this model.

Despite this knowledge, the bridge between fetal SM inflammation and mitochondrial (dys)function is yet to be established. Increasing evidence reports mitochondria, pivotal controllers of oxidative phosphorylation and energy metabolism, as key players in the underlying mechanisms of metabolic syndrome [23], obesity [24], diabetes [25], and other chronic diseases. For example, obese individuals have abnormally shaped and less efficient mitochondria, with compromised bioenergetic capacity and reduced oxidative capacity [24]. Mitochondrial density is reduced, and ATP production is compromised in T2D patients and insulin-resistant individuals [25]. In T2D, skeletal muscle mitochondrial dysfunction is observed through reduced levels of oxidative enzymes, decreased mitochondrial size, and altered mitochondrial morphology, while reduced expression of oxidative metabolism and mitochondrial respiration-related genes are reported as drivers of insulin resistance [26], supporting mitochondrial dysfunction and obesity as risk factors for insulin resistance and T2D development.

We hypothesized that obesity induced by MNE diet, before conception, and continued during pregnancy, alters fetal soleus skeletal muscle development in offspring of a non-human primate model, with consequences for mitochondrial biogenesis.

## 2. Materials and Methods

### 2.1. Care and Use of Animals

All animal procedures were approved by the Texas Biomedical Research Institute (TBRI) Institutional Animal Care and Use Committee (protocol number 1675 PC) and conducted in Association for Assessment and Accreditation of Laboratory Animal Care international-approved facilities. The experimental design adhered to the ARRIVE guidelines and incorporated the principles of the 3Rs [27]. Animals were housed in group cages with a floor area of 37 m^2^ and 3.5 m in height. Groups of up to 16 non-pregnant, outbred female baboons (*Papio* spp.) were initially housed with a vasectomized male in outdoor gang cages to promote social stability, facilitate natural social behaviors, and ensure unrestricted physical activity. Details of the housing structure and environmental enrichment have been previously published [28]. Maternal morphometric measurements were performed before pregnancy to ensure weight homogeneity and general morphometrics in females randomly assigned to the two groups, as described elsewhere [29]. Nulliparous female baboons of similar age and body weight (10–15 kg) were either assigned to the control (CTR) or maternal nutrient-excess (MNE) group. Females in the CTR group (n = 16) were fed normal chow, the Purina Monkey Diet 5038 (Purina LabDiets, St. Louis, MO, USA), containing stabilized vitamin C and all required vitamins. MNE females (n = 5) had ad libitum access to the same diet as the CTR group and additional ad libitum access to the Purina 5045-6 (Purina LabDiets, St. Louis, MO, USA) high-fat and high-energy diet, as well as free access to fructose sodas [28]. This diet was consumed for at least 12 months before pregnancy to increase body fat (Figure 1). The composition of each diet, along with the basic composition of the biscuit, is presented in Table 1. Of note, the content of protein and essential minerals and vitamins required for the baboon was matched for both CTR and MNE diets. Twelve months after the nutritional intervention, a fertile male was introduced into the group cages. Male baboons used for breeding were maintained on standard chow and were not subjected to any dietary intervention. Female baboons were monitored twice daily for general well-being and three times per week for perineal turgescence and vaginal bleeding to determine the time of conception. Pregnancy was dated initially by the changes in swelling of the sex skin and confirmed at 30 days of gestation by ultrasonography. Only singleton pregnancies were studied. 

### 2.2. Cesarean Sections and Tissue Sampling

Surgical procedures and postsurgical care were performed by a fully certified MD or DVM. Cesarean sections were performed at 165 days of gestation (90% of the gestation, 0.9 G, Figure 1) under isoflurane anesthesia (2%, 2 L/min) [28]. Fetuses were euthanized by exsanguination, removed from the uterus, and immediately submitted for morphometric analyses and tissue sampling. Fetal soleus muscle was collected, aliquoted, immediately snap frozen in liquid nitrogen, and stored at −80°. Postoperative maternal analgesia was provided with Buprenorphine hydrochloride 0.015 mg.kg^−1^.d^−1^ (Reckitt Benckiser Health Care UK Ltd., Hull, UK) during 3 days. After recovering from anesthesia, the baboon mothers were initially housed individually during the post-operative period. Following this, they were group-housed for at least 90 days with a vasectomized male to prevent pregnancy until the surgical site had fully healed [28]. 

### 2.3. Dual-Energy X-Ray Absorptiometry (DEXA)

To assess body adiposity before pregnancy, a DEXA scan was conducted in MNE (n = 5) and CTR (n = 7) females after 12 months on the respective diets. For that, a GE Lunar Prodigy 8743 (GE Healthcare, Madison, WI, USA) was used as previously described [30].

### 2.4. Immunoblotting Analysis

Immunoblotting analyses were conducted according to the procedures previously described [19]. Membranes were visualized using an Odyssey Infrared Imaging System (LI-COR Biosciences, Lincoln, NE, USA). Band density was quantified and normalized to β-tubulin protein levels. Immunoblotting was performed using the antibodies listed in Appendix A. 

### 2.5. Analysis of mtDNA Copy Number by Quantitative Real-Time PCR

Total DNA was extracted from the soleus skeletal muscle tissue using the QIAamp DNA mini-kit (#50951304 Qiagen, Düsseldorf, Germany), according to the manufacturer’s instructions. The SsoFast Eva Green Supermix (Bio-Rad, Hercules, CA, USA) was used to perform the RT-PCR, with the primers listed in Appendix A. DNA amplification was performed with an initial cycle of 2 min at 98 °C, followed by 40 cycles of 5 s at 98 °C plus 5 s at 60 °C. For quality control, melting curves were performed, and no template controls were run.

For absolute quantification and amplification efficiency, standards at known copy numbers were produced by purifying PCR products. After optimizing the annealing temperature, products were amplified for each primer pair using the HotstarTaq Master Mix Kit (#203445 Qiagen). The amplification protocol started with an initial activation step of 15 min at 95 °C, followed by 35 cycles of 1 min at 94 °C plus 1 min at 60 °C, and 1 min at 72 °C, and a final extension step of 10 min at 72 °C. After amplification, the products were purified using the MiniElute PCR purification kit (#280006 Qiagen) following the manufacturer’s instructions. All DNA was quantified using a Nanodrop 2000 device (ThermoFisher Scientific, Waltham, MA, USA), and all reactions were performed in a CFX96 real-time PCR system (Bio-Rad). mtDNA copy number was determined by the ratio between the absolute amounts of mitochondrial gene ND1 versus nuclear gene B2M, using the CFX96 Manager software (v. 3.0; Bio-Rad).

### 2.6. Real-Time Quantitative PCR (RT-PCR)

Total mRNA was extracted from the fetal soleus muscle using TRI reagent (Sigma, St. Louis, MO, USA) and reverse transcribed into cDNA using a kit (Qiagen, Valencia, CA, USA). RT-PCR was performed using the SYBR Green RT-PCR kit from Bio-Rad (Hercules, CA, USA) in a CFX96 real-time PCR system. The reaction was set with the CFX96 Manager software (v. 3.0; Bio-Rad). The list of primers is given in Appendix A. Each reaction yielded amplicons between 80–200 bp. PCR conditions were as follows: 10 s at 95 °C, 10 s at 58 °C, and 30 s at 72 °C for 40 cycles. After amplification, a melting curve (0.01 C/s) was used to confirm product purity. The changes in the threshold cycle (CT) values were calculated by the equation CT = CT target − CT input. The fold differences were calculated by the 2−(dCT) method [31]. Results are expressed relative to β-actin.

### 2.7. Measurement of Citrate Synthase and β-Hydroxyacyl-CoA Dehydrogenase Enzyme Activity

Citrate synthase and β-Hydroxyacyl-CoA dehydrogenase activities were measured by adapting the previously described protocol [32]. The detailed methodology description of each enzymatic activity is provided in the Appendix A.

### 2.8. Statistical Analysis

Each pregnant animal or the respective offspring was considered as an experimental unit. Data were analyzed as a complete randomized design using GLM (General Linear Model of Statistical Analysis System, SAS). For the offspring, the control group was composed of 8 female and 8 male fetuses, and the MNE group of 3 female and 2 male fetuses. The discrepancy of sample size between the two experimental groups resulted from the collection of additional soleus muscle samples from control groups of parallel studies conducted in the same facility, allowing a more robust analysis. Importantly, the variability and main findings remained consistent with results from smaller groups, ensuring reliability. An initial analysis of offspring birth weight and citrate synthase (CS) enzyme activity was performed to assess potential sex differences. CS activity serves as a reliable marker of mitochondrial content and oxidative capacity. Since no differences were observed between female and male fetuses (mean ± SEM: 800.5 ± 34.2 g vs. 859.0 ± 41.7 g in CTR, respectively, and 668.1 ± 64.3 g vs. 686.0 ± 13.0 g in MNE, respectively), data were pooled. Differences in mean values were compared by Tukey’s multiple comparison test, which controls for Type I error across pairwise comparisons. Mean ± standard errors of mean (SEM) are reported. Statistical significance was considered as * *p* < 0.05, ** *p* < 0.01 for MNE vs. CTR. 

## 3. Results

In this study, we evaluated the effects of a maternal nutrient-excess diet (MNE) and maternal obesity at conception on fetal soleus muscle at 0.9 G in a baboon model. Maternal obesity was induced through the MNE diet described, starting 12 months before conception and maintained during pregnancy (Figure 1).

### 3.1. Maternal and Fetal Body Weight

The MNE 12-month diet intervention increased female body weight by 18% at conception compared to females fed a normal chow diet during the same period (*p* < 0.05, Figure 2A). The DEXA scan performed at conception revealed that MNE mothers had a significant increase in overall body adiposity (*p* < 0.05, Figure 3A), especially denoted in the maternal trunk (~30% increase, Figure 3B) but also observed in the legs (~12.5%, Figure 3C) and arms (~10%, Figure 3D). CTR mothers gained on average 1.7 kg (9.4%) during gestation, while MNE mothers lost nearly 0.5 kg (2.5%) (*p* < 0.01, Figure 2C), displaying a similar body weight at cesarean section (Figure 2B). At cesarean section, fetuses of MNE mothers presented a 16% lower body weight compared to CTR fetuses (*p* < 0.01, Figure 2D).

### 3.2. MNE Increased Stress and Pro-Inflammatory Markers in Fetal Soleus Muscle

In light of the characteristic low-grade inflammation associated with obesity, we evaluated the effects of maternal obesity at conception and the MNE diet before and during pregnancy on markers related to inflammation and stress signals within the fetal soleus muscle. The transcript levels of *TNF* and *TLR4* were increased in the soleus muscle of MNE fetuses (*p* < 0.05 and *p* < 0.01, respectively, Figure 4A). A higher WB band density of the proteins encoded by these genes was also observed (*p* < 0.05, Figure 4B). Moreover, the MNE diet contributed to an increased ratio between the phosphorylated form of the NF-kB subunit p65 (Ser536) to total p65 protein content (p-p65/p65, *p* < 0.05, Figure 4C), suggesting the activation of the NF-kB pro-inflammatory pathway in the soleus muscle. A pro-inflammatory environment is further suggested by the observed activation of the p38 MAPK canonical pathway by higher levels of the phosphorylated form of p38 (Thr180/Tyr182) in fetal soleus muscle (p-p38, *p* < 0.05, Figure 4D). Pro-inflammatory signaling is usually associated with altered mitochondria function in several tissues, particularly in the context of chronic diseases, disrupting mitochondrial homeostasis and oxidative capacity [23,24]. Therefore, we next addressed how maternal obesity at conception induced by an MNE diet maintained during pregnancy affected fetal soleus muscle mitochondria quality control.

### 3.3. Mitochondrial Biogenesis and Mitochondrial Fusion Markers Are Decreased in the Soleus Muscle of MNE Fetuses

The peroxisome proliferator-activated receptors (PPAR) family plays a crucial role in cellular energy homeostasis regulation and metabolic function. In skeletal muscle, PPARs are highly involved in energy utilization and substrate preference but also play a role in inflammation. The relative mRNA levels of *PPARA PPARD* were decreased in the MNE fetal soleus muscle (*p* < 0.05, Figure 5A). Lower levels of the transcriptional co-activators of the PPARγ co-activator family, *PPARGC1A* and *PPARGC1B*, known as the “master regulators” of mitochondrial biogenesis, were also observed (*p* < 0.01 and *p* < 0.05, respectively, Figure 5A). This was accompanied by decreased cAMP responsive element binding protein 1 (*CREB1*) and endothelial nitric oxide synthase (*NOS3* transcript levels (*p* < 0.05, Figure 5C). CREB1 and NO levels, the latter partly regulated by eNOS, are highly involved in cellular signaling, contributing to the transcriptional induction of PGC1α, thus promoting mitochondrial biogenesis [33]. The protein levels of PGC1α were also found to be decreased in the MNE fetal soleus muscle *(p* < 0.05, Figure 5B). Furthermore, the transcript levels of nuclear respiratory factor 1 (*NRF1*), a crucial factor for the transcription of nuclear-encoded mitochondrial genes and whose expression is induced by PGC1α, were also lower in the soleus muscle of MNE fetuses (*p* < 0.05, Figure 5C).

Since mitochondrial quality is also governed by mitochondria’s dynamic capacity, i.e., fusion and fission processes, we assessed the levels of two transcripts involved in mitochondrial fusion. The relative mRNA expression of mitofusin 1 and 2 (*MFN1*, *MFN2*) was decreased in the MNE fetal soleus muscle *(p* < 0.05, Figure 5C), suggesting a potential impairment in mitochondrial fusion, which may have implications for mitochondrial quality control and performance.

### 3.4. Fetal Soleus Muscle Expression of SIRT1 and SIRT3

Sirtuins are deacetylases involved in the regulation of various cellular processes, including those related to mitochondrial function, biological responses to stress, metabolism modulation, and development. Sirtuins also play a protective role in several chronic diseases due to their distinct ability to regulate autophagy. While SIRT1 has a diverse subcellular localization pattern, SIRT3 is primarily localized in mitochondria [34,35]. In this study, *SIRT1* and *SIRT3* transcript levels were both lower in the soleus muscle of MNE fetuses (*p* < 0.05, Figure 6A) and while only a decreasing trend was observed for SIRT1 protein levels (Figure 6B), SIRT3 levels were lower in MNE fetal soleus muscle (*p* < 0.05, Figure 6B), with possible implications for mitochondrial oxidative capacity and function regulation.

### 3.5. Decreased Fetal Skeletal Muscle Mitochondrial Abundance and Function

Diverse parameters can be used to assess mitochondrial health in frozen tissues. In this work, we determined mitochondrial DNA copy number, often used as a direct indicator of mitochondrial content. In the soleus muscle of MNE fetuses, mtDNA copy number was decreased compared to the levels observed in CTR fetuses (*p* < 0.05, Figure 7A). Alterations in transcripts and protein levels can often manifest as alterations in enzymatic activities. Citrate synthase’s (CS) activity, commonly used as a marker to assess mitochondrial content, showed a decreasing trend in the soleus muscle of MNE fetuses (Figure 7B). A similar trend was observed in the activity of a key mitochondrial enzyme involved in the third reaction of β-oxidation, the β-hydroxyacyl-CoA dehydrogenase (BHAD, Figure 7B). Elements of the mitochondrial electron transport chain (ETC) might also be subject to dysregulation due to chronic disease. In the fetuses of MNE mothers, the protein expression of cytochrome C was lower (*p* < 0.05, Figure 7C). Cytochrome C is an ETC essential component that transfers electrons between ubiquinol cytochrome C oxidoreductase (Complex III) and cytochrome C oxidase (Complex IV), the latter being the terminal enzyme of the mitochondrial respiratory chain, as it catalyzes the electron transfer from reduced cytochrome C to molecular oxygen. Three mitochondrial DNA-encoded subunits compose the cytochrome C oxidase catalytic core (COX I, COX II, and COX III). The transcript levels of the I*cox1* and *cox2* subunits were decreased in the soleus skeletal muscle of MNE fetuses (*p* < 0.05, Figure 7D). All in all, these data suggest impaired mitochondrial function in the soleus muscle of fetuses from mothers obese at conception and fed an MNE diet throughout pregnancy.

## 4. Discussion

Maternal obesity and nutrient-excess diets are increasingly common and can disrupt fetal metabolism, with lasting effects on offspring health [36]. While adverse intrauterine environments are linked to chronic disease risk, the molecular mechanisms involved remain unclear. Using a non-human primate model (baboon), we show that maternal metabolic stress activates pro-inflammatory and stress factors in the fetal soleus skeletal muscle, affecting mitochondrial content and activity. 

Consumption of an MNE diet at least 12 months pre-conception resulted in increased pre-pregnancy body weight in female baboons and increased fat mass, especially in the trunk, denoting a dysfunctional fat distribution. Obesity in humans also portrays an abnormal fat distribution, predominantly in the upper body, with increased visceral fat accumulation [37], which is associated with an increased risk of insulin resistance, dyslipidemia, hypertension, and other irregular metabolic factors observed within the metabolic syndrome cluster [38] and that predispose for T2D and other chronic diseases [39].

In the current study, female baboons on the MNE diet presented reduced gestational weight gain (GWG). Women with pre-pregnancy overweight or obesity have the highest prevalence of excessive GWG [11]. Thus, studies have highly focused on how excessive GWG on obese mothers predisposes them and their offspring to complications (e.g., macrosomia) [40,41]. Some studies suggest that pre-pregnancy weight has a stronger impact on childhood overweight/obesity [8], while emerging evidence links GWG below the guidelines in obese mothers to low birth weight and other offspring complications [42,43], even independently of pre-pregnancy BMI [44]. As previously shown for our model, despite the reduced GWG, MNE mothers show potential metabolic disturbances with increased circulating levels of low-density lipoprotein and triglycerides during pregnancy compared to CTR mothers. The increased triglycerides were still observed at cesarean section and accompanied by increased levels of glucose and insulin [45]. A meta-analysis including only obese women showed that mothers with inadequate GWG had a higher risk of having small for gestational age babies [43]. Indeed, in our study, fetuses of MNE baboons had decreased body weight at term, which could be a consequence of the observed low maternal GWG. A compelling explanation for the observed effects could be the decreased placental efficiency, previously reported in this model [46]. Similar effects were observed in offspring of high-fat fed female mice [47,48]. Our model also previously showed decreased fetal blood levels of amino acids directly involved in one-carbon metabolism, accompanied by a higher Apolipoprotein B to Apolipoprotein A1 ratio, an established marker of atherogenic and cardiovascular risk [46]. Moreover, circulating fetal insulin levels were increased at cesarean section [45]. On another note, a human investigation that included 905 mother-child pairs enrolled in the Hyperglycemia and Adverse Pregnancy Outcome (HAPO) study revealed that independently of pre-pregnancy BMI, both excessive and inadequate GWG were associated with increased offspring hypertension and insulin resistance at 7 years of age [44]. Altered postnatal growth trajectories have already been reported in our model, showing that offspring of obese mothers, with reduced body weight at the fetal stage, had a steeper growth in their first 2 years of life [49].

The maternal obesity-promoted low-grade inflammation generates an intrauterine environment suboptimal for offspring health. One well-known pro-inflammatory cytokine is TNFα, which has been highly associated with obesity and T2D given its implications for insulin signaling and insulin resistance development [50]. Although mostly known as an adipokine, TNFα is also produced in SM [51]. Both transcript and protein TNFα levels were increased in the SM of MNE fetuses, and these were accompanied by increased levels of other proteins involved in inflammatory pathways. We previously showed that TLR4, a key receptor abundantly expressed in the SM and highly involved in immune and inflammatory responses, was upregulated in the sheep fetal SM [19]. Microbial lipopolysaccharides are the main agonists of TLR4, but saturated fatty acids and fibrinogen have been suggested as stimuli for TLR4 dimerization and signaling cascade activation in the context of obesity [21,52]. TLR4 signaling cascade can, through a series of interactions, lead to NF-kB and MAPK activation, ultimately resulting in the transcription of pro-inflammatory factors [52], including TNFα. In this study, both transcript and protein levels of TLR4 were higher in MNE baboon fetal SM, which was accompanied by increased phosphorylation of the NF-kB p65 heterodimer, suggesting activation of the NF-kB pathway, in accordance with our observations in ewes [19]. The increased phosphorylation of p38 MAPK observed further supports this TLR4-induced inflammatory response, given that p38 MAPK can upregulate activator protein 1 (AP1), resulting in pro-inflammatory cytokines transcription.

Nuclear factor-kB activation has been associated with mitochondrial dysfunction in the muscle in response to cellular fuel overload in L6 myotubes, with decreased PGC1α expression, as well as in other genes related to mitochondrial dynamics [53], a similar effect to the one observed in the MNE baboon fetal SM in this study. The group of transcription factors PPAR (PPARα, PPARδ, and PPARγ) has an important role in the SM’s metabolic flexibility but also in the regulation of inflammatory responses [54]. Particularly, PPARδ is highly expressed in the SM [55] and has been shown to stimulate PGC1α expression in mice SM, while its ablation induced muscle fiber-type switching, reducing muscle oxidative capacity and consequently predisposing to obesity and diabetes [56]. Indeed, the transcript levels of *PPARB* were lower in the SM of fetal MNE baboon in our study and coupled to the observed reduced transcript levels of *PPARA*, another transcription factor involved in the regulation of lipid metabolism and fatty acid uptake and the second most expressed in the SM [57], these data suggest an impairment in the fetal SM metabolic flexibility, which could have repercussions for MNE offspring future health. Peroxisome proliferator-activated receptors have also been shown to regulate inflammation processes in the SM, and, for instance, PPARα has been shown to inhibit NF-kB transcriptional activity [54].

Several genes involved in mitochondrial dynamics and homeostasis are regulated by PGC1α and PGC1β. Studies have associated mitochondrial alterations in the SM of T2D patients and animal models with altered PGC1α, PGC1β, and Mfn2 levels [58]. In the current study, *PPARGC1A* and *PPARGC1B* transcript levels were reduced, which was previously found in the human SM of not only T2D patients but also of offspring from insulin-resistant individuals [58]. Decreased *PPARGC1B* mRNA levels were also observed in the SM of pig fetuses of high-energy fed mothers [59] and in mice fetal SM of hyperglycemic mothers at late gestation [60]. Moreover, lower levels of the MFN2 and MFN1 transcripts were also observed, suggesting unbalanced mitochondrial dynamics, with impaired fusion events. Multiple studies have reported reduced MFN2 levels in the context of obesity, diabetes, and high-fat diets in the SM, and modulation of its expression can reverse, prevent, or ameliorate SM mitochondrial function [61]. Although these effects do not seem to regularly extend to alterations in MFN1 levels [61], in our study, MFN1 transcript levels were more strongly impacted in the fetal SM of MNE offspring than MFN2 levels, which may represent a specific response of fetal SM programming by maternal obesity at conception and nutrient excess during pregnancy.

The transcription of PGC1α is enhanced by the interaction of multiple factors with the promoter region of its gene. The transcription factor CREB is capable of promoting PGC1α transcription, although mostly studied in response to exercise [62]. Palacios et al. showed that not only exercise, but also nutritional status can influence PGC1α transcription through SIRT3 levels [34]. The study revealed decreased SIRT3 levels due to a high-fat diet and that SIRT3 knockout resulted in decreased CREB phosphorylation and PGC1α transcription [34]. Yan and colleagues reported decreased CREB phosphorylation associated with reduced *Ppargc1α* transcription in the SM of mice fetuses exposed to a hyperglycemic intrauterine environment [60]. In the present study, SIRT3 transcript and protein levels were decreased in the fetal SM of MNE offspring. Although we only measured *CREB1* transcript levels, which were lower in the SM of MNE fetuses, regulation of PGC1α transcription through CREB interaction modulated by diet deserves further attention, especially since it is well described that SIRT3 activity is increased in response to exercise and poor nutrient intake and decreased in diabetic models [35]. Likewise, SIRT1 stimulates PGC1α [63], and exercise training contributes to increased activation of the SIRT1/PGC1α axis with reduced NF-kB signaling in diabetic mice [64], demonstrating a crosstalk between inflammatory mediators and PGC1α. One of the observed effects of PGC1α transcription enhancement by these factors is the upregulation of NRF1 and mitochondrial transcription factor A (TFAM), required for mitochondrial DNA transcription [64]. In this study, the lower *CREB1*, *SIRT1*, and *SIRT3* transcript levels alongside decreased *PPARGC1A* and *NRF1* suggest dysregulation of mitochondrial biogenesis in the SM of fetuses developed in a nutrient-excess environment. Zou et al. reported decreased transcription of *NRF1*, *TFAM,* and *SIRT1* in fetal SM of pigs exposed to maternal high-energy intake, with decreased mRNA levels of DNA polymerase (POLG) and single-strand DNA binding protein 1 (SSBP1), accompanied by lower mtDNA copy number [59]. McCurdy et al. also reported reduced *PGC1α* expression in fetal SM of Japanese macaques exposed to a maternal Western-style diet and obesity. Although *TFAM* and *MFN2* transcript levels were not significantly altered, they observed a reduction in mitochondrial content [65]. In the current study, mtDNA copy number was also found to be decreased in the SM of MNE baboon fetuses. Further supporting the involvement of inflammatory cues in mitochondrial biogenesis dysregulation, a study revealed that lower mRNA expression of *Ppargc1a*, *Nrf1,* and *Tfam,* accompanied by lower expression of eNOS, was ameliorated by TNFα signaling deficiency in the SM of different rodent models of obesity [66]. Accordingly, the transcript levels of *NOS3* in the fetal SM of MNE baboons were substantially lower. On another note, increased NO levels, partly derived from increased eNOS, can lead to increased S-nitrosylation of CREB, stimulating its binding to the PGC1α gene promoter, thus increasing PGC1α transcription [33]. 

Unsurprisingly, some of the studies mentioned above reported that alterations in mitochondrial-related proteins were important for SM mitochondrial oxidative capacity. For instance, in Liu and Chang’s work, cytochrome C oxidase activity was lower in diabetic mice, and exercise stimulation contributed to an increase in mRNA levels of mitochondrial complexes subunits, through the induction of PGC1α, NRF1, and TFAM levels [64]. Moreover, Valerio et al. reported decreased protein levels of COX IV and cytochrome C in obese rodents with decreased eNOS expression [66]. Cytochrome C protein levels were lower in the fetal SM of MNE offspring in our study, and we also found implications for COX IV catalytic core subunits COX I and COX II, with lower mDNA copy number, lower mitochondrial CS and BHAD enzymatic activity. In the previously mentioned study using fetal SM from Japanese macaques, mitochondrial electron transport chain activity was elevated, but coupling efficiency was reduced. This was accompanied by lower CS activity and increased markers of oxidative stress [65]. Levels of mtDNA were also reported to be lower in the different rodent models of obesity [66].

Overall, our data show a strong association between maternal obesity at conception and a nutrient-excess diet during pregnancy-induced chronic inflammation in the fetal soleus skeletal muscle, with implications for mitochondrial content and function. Furthermore, PGC1α seems to represent a major interface between inflammatory signaling pathways and SM mitochondrial alterations that are already present at late gestation in a non-human primate model. 

## 5. Conclusions

The underlying mechanisms of the adverse effects of maternal nutrition and health on fetal soleus skeletal muscle development and metabolism are still poorly understood, especially with regard to mitochondrial consequences. Despite the constant emphasis on chronic inflammation induced by maternal obesity or diabetes, there is a lack of molecular studies during the fetal stage and in animal models with high human translational relevance. In this study, we demonstrate that baboon fetuses exposed to an adverse intrauterine environment resulting from maternal obesity at conception and a nutrient-excessive diet during pregnancy exhibit molecular alterations in the soleus muscle. Specifically, we observed altered activity in soleus muscle mitochondria, posing a potential risk for widespread cellular dysfunction and early disease development. Notably, these effects could potentially program the fetus for insulin resistance, a key factor in the development of T2D, thereby increasing the risk of life-long metabolic complications in the postnatal period.

## 6. Study Limitations

In this study, we only measured the transcript levels of *CREB1*, *NOS3*, *MFN1*, *MFN2*, *NRF1*, *cox1*, and *cox2*. While we observed differences between the groups, it is crucial to recognize that mRNA levels may not accurately reflect protein content or enzymatic activity. For instance, evaluating the activity of CREB1 and eNOS could have provided a more comprehensive understanding of their contributions to the alterations observed in this study, particularly regarding mitochondrial biogenesis, and might have identified alternative pathways worthy of exploration. One limitation of our NHP model, consistent with observations in other NHP studies, is that despite its robust translational value, the small sample size restricted our capability to assess sexual dimorphism. Nevertheless, when dealing with maternal interventions, acknowledging sex-specific responses is imperative in cohorts with larger sample numbers. This is crucial not only because morphometric parameters differ but also due to variations in molecular alterations observed in different tissues based on fetal sex [67,68,69]. Furthermore, it is important to note that we exclusively used the first cohort of animals, consisting of only 5 MO offspring, paired in time with the respective controls. This limited sample size, along with the potential for individual variation, further restricted our ability to explore more granular aspects of the data, such as sexual dimorphism or the potential for other molecular alterations not captured in this initial analysis. 

## Figures and Tables

**Figure 1 biology-14-00868-f001:**
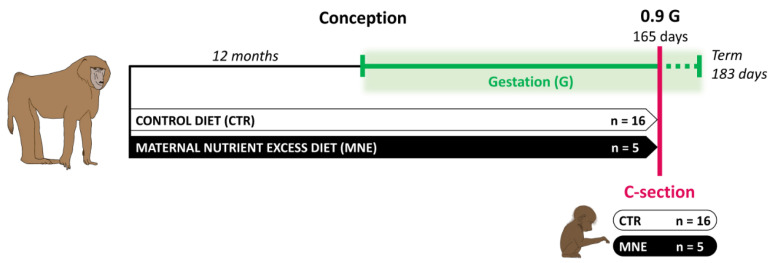
Maternal diet before conception and during pregnancy until cesarean section (C-section) at 90% of the gestation (0.9 G). CTR—control diet; MNE—maternal nutrient-excess diet.

**Figure 2 biology-14-00868-f002:**
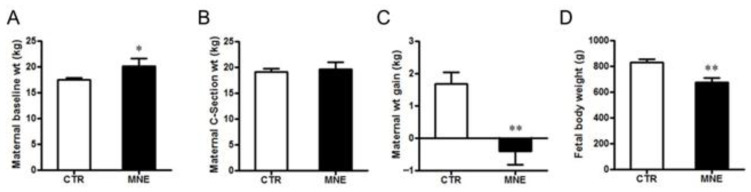
Body weight of mothers subjected to a control (CTR) or maternal nutrient-excess (MNE) diet, and respective fetuses developed in these different maternal conditions. (**A**) Maternal body weight at conception (baseline). (**B**) Maternal body weight at cesarean section (90% of the gestation). (**C**) Maternal weight gain during pregnancy. (**D**) Fetal body weight at cesarean section. CTR: open, n = 16; MNE: closed, n = 5. Data are expressed as mean ± SEM. ** *p* < 0.01, * *p* < 0.05.

**Figure 3 biology-14-00868-f003:**
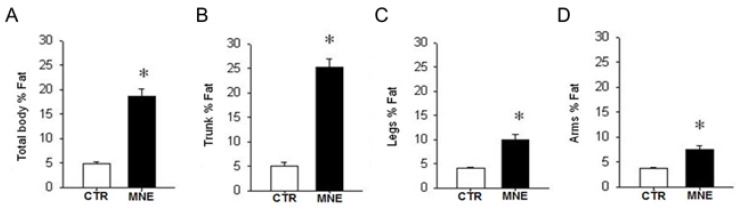
Maternal fat percentage from DEXA scan evaluation after 12 months on the control (CTR) or maternal nutrient-excess (MNE) diet, before conception. (**A**) Total body fat percentage. Fat percentage in the (**B**) trunk, (**C**) legs, and (**D**) arms. CTR: white, n = 7; MNE: black, n = 5. Data are expressed as mean ± SEM; * *p* < 0.05.

**Figure 4 biology-14-00868-f004:**
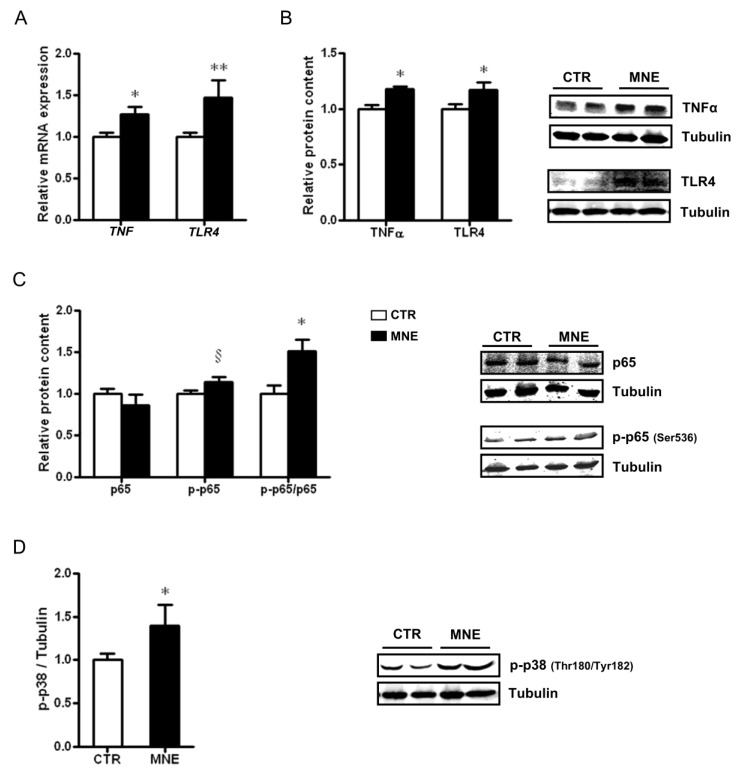
Markers involved in pro-inflammatory pathways and stress responses in fetal baboon soleus muscle at 90% of the gestation were exposed to different maternal conditions, including control (CTR) diet and maternal nutrient-excess (MNE) diet, which included a high-fat, high-sugar diet, and free access to high-fructose sodas. (**A**) Transcript levels of tumor necrosis factor α (*TNF*) and toll-like receptor 4 (*TLR4*). (**B**) Relative protein content of TNFα and TLR4. (**C**) Nuclear factor kappa-light-chain-enhancer of activated B cells (NF-κB) subunit p65, phospho-p65 protein level, and ratio of phosphorylated p65 to total protein. (**D**) Protein levels of phospho-p38 mitogen-activated protein kinase (MAPK). CTR: white, n = 16; MNE: black, n = 5. Data are expressed as mean ± SEM. ** *p* < 0.01, * *p* < 0.05, § *p* < 0.10.

**Figure 5 biology-14-00868-f005:**
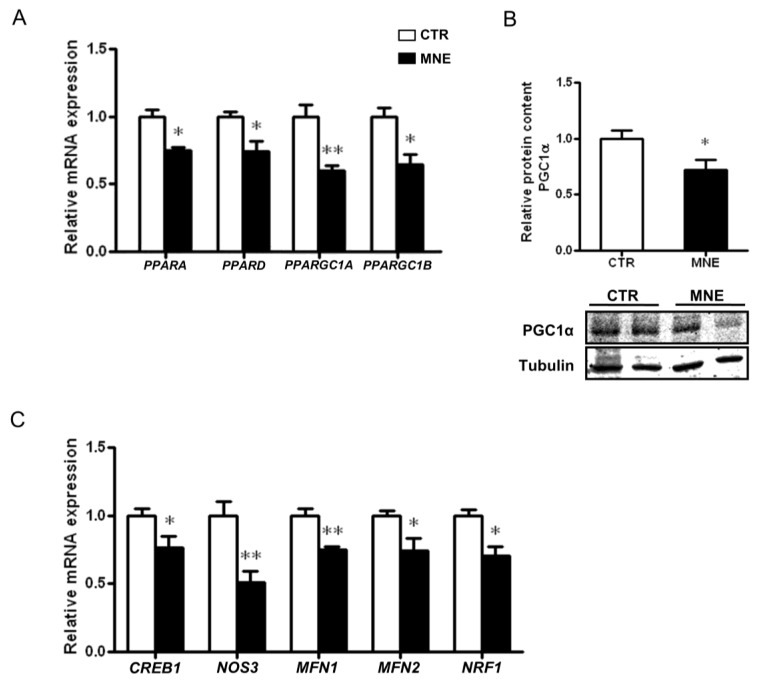
Effects of maternal nutrient excess (MNE) versus control (CTR) conditions on mitochondrial biogenesis and fusion-related markers in the fetal soleus muscle at 90% of the gestation. (**A**) Transcript levels of peroxisome proliferator-activated receptor (PPAR) α (*PPARA*) and *PPARD* and mitochondrial biogenesis-related genes PPARγ coactivator 1 α (*PPARGC1A*) and *PARGC1B*. (**B**) Relative protein levels of PGC1α. (**C**) Transcript levels of cAMP responsive element binding protein 1 (*CREB1*), endothelial nitric oxide synthase (*NOS3*), mitofusin 1 (*MFN1*), *MFN2,* and nuclear respiratory factor 1 (*NRF1*); CTR: white, n = 16; MNE: black, n = 5. Data are expressed as mean ± SEM. ** *p* < 0.01, * *p* < 0.05.

**Figure 6 biology-14-00868-f006:**
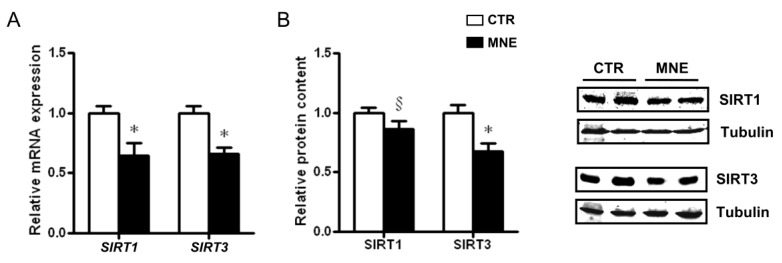
Fetal soleus muscle expression of SIRT1 and SIRT3 at 90% of the gestation as a result of control (CTR) or maternal nutrient-excess (MNE) diets. (**A**) Transcript levels of sirtuin 1 (*SIRT1*) and *SIRT3*. (**B**) Relative protein content of SIRT1 and SIRT3. CTR: white, n = 16; MNE: black, n = 5. Data are expressed as mean ± SEM. * *p* < 0.05, § *p* < 0.10.

**Figure 7 biology-14-00868-f007:**
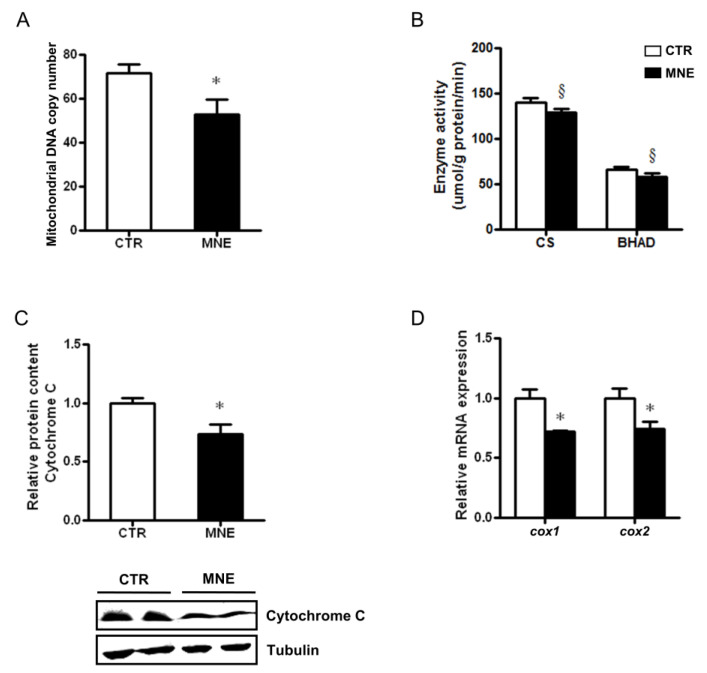
Alterations in fetal soleus muscle mitochondrial content at 90% gestation in mothers on control (CTR) or maternal nutrient-excess (MNE) diets. (**A**) Mitochondrial DNA copy number. (**B**) Enzymatic activities of citrate synthase (CS) and β-hydroxyacyl-CoA dehydrogenase (BHAD). (**C**) Relative protein content of cytochrome C. (**D**) Transcript levels of cytochrome C oxidase subunit 1 (*cox1*) and 2 (*cox2*). Control (CTR): white, n = 16; Maternal nutrient excess (MNE): black, n = 5. Data are expressed as mean ± SEM. * *p* < 0.05, § *p* < 0.10.

**Table 1 biology-14-00868-t001:** Composition of experimental diets and basic biscuit formulation.

Energy Source (%)	Purina Monkey Diet 5038	Purina 5045-6
Fat	12	45
Glucose	0.29	4.62
Fructose	0.32	5.64
**Metabolizable Energy Content (kcal/g)**	3.07	4.05
**Biscuit Basic Composition**	Crude protein (≥15%), crude fat (≥5%), crude fiber (≤6%), ash (≤5%), added minerals (≤3%)	

## Data Availability

The data presented in this study are available on request from the corresponding author.

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
