# Peer review of "Maternal Nutrient Excess Induces Stress Signaling and Decreases Mitochondrial Number in Term Fetal Baboon Skeletal Muscle"

_biology, 2025, doi:10.3390/biology14070868_

Round 1

Reviewer 1 Report

Comments and Suggestions for Authors

In this manuscript, the authors tried to explain the role of maternal nutrient excess through the Baboon as a fetal programming model. However, certain formatting and language issues need further improvement. The major comments and suggestions needed to be improved in this draft are listed below:

Comment 1: The authors forgot to attach the supplementary data while submission, so it is advised to include supplementary data files when resubmitting the updated draft.

Comment 2: The references, either in-text citations or in the form bibliography, at the end of this draft are not formatted as per the guidelines of Biology.

Comment 3: All the gene symbols, especially when highlighting the transcript expression, should be italicized throughout the manuscript as examples mentioned at lines 34-35 of the abstract and in lines 531-532 of the conclusion. The whole document should be updated accordingly.

Comment 4: Lines 39-41 should be restructured and reworded.

Comment 5: The surveys and reports should be properly cited throughout the draft, as an example, see lines 48-54 and 55-57, as there are no proper citations for the facts mentioned here.

Comment 6: The in-text citations, especially at lines 60, 63, 70, 76, and throughout the documents, should be double-checked to align with Biology guidelines.

Comment 7: The lines 87-88 should be restructured and reworded for better understanding.

Comment 8: The facts mentioned on lines 87-88 should also be included if the same model of maternal nutrition was under consideration here.

Comment 9: Lines 93-95 and 96-99 lack proper citation.

Comment 10: Lines 104-105 and 113-115 need proper rewording and restriction for better understanding.

Comment 11: Lines 120-123 lack a citation for the guidelines mentioned here.
Comment 12: Lines 130-131 and 133-137 should be restructured, and feed treatments should be summarized in a table format.

Comment 13: The company details for feed treatments on lines 130 and 135, along with details for brands mentioned on lines 151-152, should be reformatted.

Comment 14: Lines 163-172 and 176-180 should be restructured to a tabulated form for better understanding. Moreover, the primer details for each gene should be in tabulated form.

Comment 15: Lines 203-205 do not have any citation for the methods used.

Comment 16: Line 223, as mentioned the use of Tukey adjustment, then instead of p-value, the adj. p-value and FDR values should be reported for all the results in the manuscript.

Comment 17: All figures from Figure 4 onward show a graph for genes and their expression levels, but the treatment groups are properly highlighted in the figures as well as in the legends. So, these should be updated accordingly.

Comment 18: Lines 268-270 should be cited properly and lack citation(s).

Comment 19: Lines 298-301 should be restructured for better understanding.

Comment 20: Lines 333-337 should be included in the discussion rather than here in the results.

Comment 21: Lines 364-378 in the discussion are more like an introduction to this draft and should be summarized in 1-2 lines to move forward with the discussion.

Comment 22: Line 430, mentioning a study on ewes lacks a reference citation.

Comments 23: Sentences started with acronyms or gene names should be restructured like in lines 434, 447, and should be updated throughout the document.

Comment 23: Lines 483-484 mentioned the previous related study facts and lack proper citation.

Comments on the Quality of English Language

Some sentences, as mentioned in the author's comments, need restructuring and rewording. Moreover, the references should be properly formatted as well according to the guidelines of Biology.

Author Response

REVIEW 1 (R1)
Comments and Suggestions for Authors

R1: In this manuscript, the authors tried to explain the role of maternal nutrient excess through the Baboon as a fetal programming model. However, certain formatting and language issues need further improvement. The major comments and suggestions needed to be improved in this draft are listed below:

Answers authors (A): We thank the reviewer for their thorough evaluation of our manuscript and constructive feedback. We appreciate the recognition of our work. We have addressed the formatting and language issues to improve the manuscript’s overall presentation. Our detailed responses to each comment are provided below.

R1: Comment 1: The authors forgot to attach the supplementary data while submission, so it is advised to include supplementary data files when resubmitting the updated draft.

A: Thank you for noting this. The supplementary data file will be included with the resubmission of the updated manuscript.

R1: Comment 2: The references, either in-text citations or in the form bibliography, at the end of this draft are not formatted as per the guidelines of Biology.

A: Thank you for your observation. The references, including in-text citations and the bibliography, have been reformatted to comply with the guidelines of Biology.

R1: Comment 3: All the gene symbols, especially when highlighting the transcript expression, should be italicized throughout the manuscript as examples mentioned at lines 34-35 of the abstract and in lines 531-532 of the conclusion. The whole document should be updated accordingly.

A: We acknowledge this important point and have addressed it as follows. All gene symbols have been italicized throughout the manuscript, particularly in cases referring to transcript expression, as noted in the abstract and conclusion. In addition, we ensured that gene and protein representations are appropriately distinguished and corrected any inconsistencies where gene symbols were incorrectly used in place of protein names. Figures and figure legends were also updated accordingly to reflect these changes, in the manuscript and supplementary data material.

R1: Comment 4: Lines 39-41 should be restructured and reworded.

A: The final sentence of the abstract was revised to more clearly convey the intended message.

R1: Comment 5: The surveys and reports should be properly cited throughout the draft, as an example, see lines 48-54 and 55-57, as there are no proper citations for the facts mentioned here.

A: Thank you for your comment. The surveys and reports have now been appropriately cited within the manuscript, and the referenced information has been updated to incorporate the most recent data available.

R1: Comment 6: The in-text citations, especially at lines 60, 63, 70, 76, and throughout the documents, should be double-checked to align with Biology guidelines.

A: The references have been formatted according to the guides of Biology.

R1: Comment 7: The lines 87-88 should be restructured and reworded for better understanding.

A: The sentence in the mentioned lines has been restructured and reworded to improve clarity and readability.

R1: Comment 8: The facts mentioned on lines 87-88 should also be included if the same model of maternal nutrition was under consideration here.

A: Thank you for your comment. We confirm that a similar model of maternal nutrient excess was used in the referenced study on ewes, with nutrient excess initiated prior to pregnancy, consistent with the NHP model used in the current study. To clarify this continuity for the reader, we have revised the text in the manuscript to explicitly state that both models involved maternal nutrient excess exposure before and during pregnancy.

R1: Comment 9: Lines 93-95 and 96-99 lack proper citation.

A: Citations were added to the mentioned sections of the manuscript.

R1: Comment 10: Lines 104-105 and 113-115 need proper rewording and restriction for better understanding.

A: The sentences in the mentioned lines were rephrased to improve clarification.

R1: Comment 11: Lines 120-123 lack a citation for the guidelines mentioned here.

A: A citation for the guidelines was added to the text.

R1: Comment 12: Lines 130-131 and 133-137 should be restructured, and feed treatments should be summarized in a table format.

A: Thank you for the helpful suggestion. The mentioned lines have been restructured, primarily by removing most of the diet information from the text, which is now summarized in a new table (Table 1) to provide a clearer and more accessible overview of each diet’s composition.

R1: Comment 13: The company details for feed treatments on lines 130 and 135, along with details for brands mentioned on lines 151-152, should be reformatted.

A: The company and brand details mentioned have been reformatted according to the journal’s guidelines (Company, City, Country).

R1: Comment 14: Lines 163-172 and 176-180 should be restructured to a tabulated form for better understanding. Moreover, the primer details for each gene should be in tabulated form.

A: The information mentioned was added to a table in the supplementary material document (as Table S1, “List of primary and secondary antibodies used for immunoblotting” and S2, “List of primers used for transcript amplification”).

R1: Comment 15: Lines 203-205 do not have any citation for the methods used.

A: A citation for the methods used for the calculation of the CT value was added to the manuscript.

R1: Comment 16: Line 223, as mentioned the use of Tukey adjustment, then instead of p-value, the adj. p-value and FDR values should be reported for all the results in the manuscript.

A: We acknowledge the importance of multiple testing corrections such as FDR. In this study, we applied the Tukey adjustment, which is appropriate for controlling the risk of Type I error across multiple pairwise comparisons. Accordingly, we have reported the Tukey-adjusted p-values throughout the manuscript. While FDR is typically used in settings involving a large number of comparisons, the scope and design of our analyses were better suited to this type of pairwise multiple comparison correction. We have clarified this in the revised text and appreciate your understanding.

R1: Comment 17: All figures from Figure 4 onward show a graph for genes and their expression levels, but the treatment groups are properly highlighted in the figures as well as in the legends. So, these should be updated accordingly.

A: Thank you for the observation. All figures from Figure 4 onward have been reviewed. Group labels were updated where necessary to enhance the visual distinction between treatment groups, and the figure legends were clarified accordingly. To improve clarity, the terms "open" and "closed" for the CTR and MNE groups, respectively, were replaced with "white" and "black." These changes have been applied consistently across all figures.

R1: Comment 18: Lines 268-270 should be cited properly and lack citation(s).

A: Thank you for your comment. Appropriate citations have been added to the mentioned sentence to support the information presented.

R1: Comment 19: Lines 298-301 should be restructured for better understanding.

The sentence mentioned was split and restructured to improve clarity and readability.

R1: Comment 20: Lines 333-337 should be included in the discussion rather than here in the results.

A: Thank you for your suggestion. We agree that interpretation is typically done in the discussion section, however, we believe that the brief contextualization provided is important to guide the reader’s understanding of the results. Nevertheless, we made some changes to the manuscript’s text. The sentence “Mitochondrial enzymatic activities are often related to metabolic disturbances” was deleted, and the following sentence was adjusted to sound more descriptive.

R1: Comment 21: Lines 364-378 in the discussion are more like an introduction to this draft and should be summarized in 1-2 lines to move forward with the discussion.

A: We appreciate the reviewer’s comment. We have revised this section to present a more concise and focused summary of the key ideas, improving clarity and readability.

R1: Comment 22: Line 430, mentioning a study on ewes lacks a reference citation.

A: A reference has now been added to support the statement regarding the study on ewes.

R1: Comments 23: Sentences started with acronyms or gene names should be restructured like in lines 434, 447, and should be updated throughout the document.

A: Thank you for pointing this out. We have revised the document and the paragraphs and sentences starting with acronyms or abbreviations were restructured throughout the document.

R1: Comment 23: Lines 483-484 mentioned the previous related study facts and lack proper citation.

A: The proper citation was added to the sentence mentioned.

Comments on the Quality of English Language

Some sentences, as mentioned in the author's comments, need restructuring and rewording. Moreover, the references should be properly formatted as well according to the guidelines of Biology.

We have thoroughly restructured and reworded the relevant sentences throughout the manuscript. Additionally, we have carefully reviewed and revised all references to ensure they conform to the formatting requirements of Biology

Reviewer 2 Report

Comments and Suggestions for Authors

Review Report

This study is based on a non-human primate model, which, although limited by a small sample size, holds significant translational value within the DOHaD field. With appropriate justification of the experimental design, such models can provide highly relevant insights for the scientific community, especially in fields where rodent models may not fully capture human physiology.

The manuscript provides valuable data on the effects of maternal nutrient excess and its developmental consequences. I recommend a minor revision to improve clarity and consistency. Suggestions include refining figure labels, clarifying statistical methods for small sample sizes, defining key terms and acronyms, and briefly expanding on relevant physiological and translational aspects. These adjustments will enhance the overall readability and scientific rigor of the manuscript.

Line 219: Since the sample size is only two or three for some groups/sexes, it is important to clarify how birth weight was statistically compared and how the standard error of the mean (SEM) was estimated.

Line 221: It would be important to specify which physiological parameters were included in the initial analysis mentioned.

Line 258: Consider adding the phrase "before pregnancy" after "MNE diet" to emphasize the experimental procedure, in which maternal females were fed the MNE diet at least 12 months prior to pregnancy.

Line 305: It would be helpful to include the p-value in parentheses before "Figure 6B" to illustrate the strength of the described trend.

Lines 308, 319, and 351: In the corresponding figure panels, consider adding the appropriate labels below each bar for the CTR and MNE groups, as done in Figures 2 and 3. Additionally, including the measured parameter centered below the CTR and MNE bars could improve clarity and consistency.

Line 335: If applicable, include the abbreviation in parentheses immediately after "Citrate synthase."x

Line 356: In Figure 7C, consider including the term "Mitochondrial DNA" in the axis title, if applicable, to provide a clearer description of the results. Additionally, ensure consistency in axis labeling for relative protein content with respect to the labeling used in Figure 5 for example. Including group labels (CTR and MNE) in the bar plots, where missing, would also improve clarity.

Line 373: It would be more informative if the authors discussed the developmental differences at birth between rodents and non-human primates, rather than broadly stating that rodents have low translational value. Rodent models have provided valuable physiological and molecular insights within the DOHaD field, and acknowledging this would offer a more balanced perspective.

In this part of the discussion, it would be valuable to cite relevant findings from rodent models that are consistent with intrauterine growth restriction observed in the context of maternal obesity or maternal nutrient excess. Including such references would strengthen the discussion and highlight the translational relevance of these experimental models.

Lines 405–406: It would be useful to very briefly mention the physiological significance of the ApoB/ApoA1 ratio to help contextualize its relevance within the study’s findings.

Line 407: It would be helpful for readers if the acronym "HAPO" were defined when first mentioned, to ensure clarity and accessibility for a broader audience

Author Response

REVIEW 2 (R2)

Comments and Suggestions for Authors

Review Report

R2: This study is based on a non-human primate model, which, although limited by a small sample size, holds significant translational value within the DOHaD field. With appropriate justification of the experimental design, such models can provide highly relevant insights for the scientific community, especially in fields where rodent models may not fully capture human physiology.

The manuscript provides valuable data on the effects of maternal nutrient excess and its developmental consequences. I recommend a minor revision to improve clarity and consistency. Suggestions include refining figure labels, clarifying statistical methods for small sample sizes, defining key terms and acronyms, and briefly expanding on relevant physiological and translational aspects. These adjustments will enhance the overall readability and scientific rigor of the manuscript.

A: We thank the reviewer for their constructive comments. We appreciate the recognition of the translational value of our non-human primate model and the importance of our findings to the DOHaD field. In response to the suggestions, we have revised the manuscript to improve clarity and consistency. We hope the updated version addresses the reviewer’s concerns appropriately.

R2: Line 219: Since the sample size is only two or three for some groups/sexes, it is important to clarify how birth weight was statistically compared and how the standard error of the mean (SEM) was estimated.

A: Thank you for highlighting this important point. We agree that the small sample size in certain groups requires clarification. The standard error of the mean (SEM) was calculated using the conventional formula (SEM = SD / √n). However, we acknowledge that, due to the limited number of animals, the SEM should be interpreted with caution and is provided here for descriptive purposes only.

Regarding the sex distribution of the HF-HED fetuses (n = 5), three were female and two male. Given this unbalanced and small sample size, we did not perform sex-based analyses. This decision aligns with previous studies, such as the one by Nathanielsz et al. 2015 ( PMID: 26537341), which also refrained from sex-specific analysis under similar constraints. Furthermore, as noted in another relevant study (PMID: 29516496), no significant sex differences were observed, and the authors therefore pooled male and female data within each dietary treatment group for their analysis.

R2: Line 221: It would be important to specify which physiological parameters were included in the initial analysis mentioned.

A: We appreciate your comment and acknowledge that our original phrasing regarding sex differences may have lacked clarity. We have revised the relevant sentence to better reflect our rationale. Specifically, offspring birth weight and citrate synthase (CS) enzyme activity were initially analyzed to explore potential sex differences. However, due to the small sample size and the absence of consistent differences between sexes, the data were pooled by treatment group (CTR or MNE) for subsequent analysis.

Sex differences in skeletal muscle metabolism have been partly attributed to variations in mitochondrial content and function. So we selected citrate synthase activity, a stable and reproducible marker of mitochondrial abundance and oxidative capacity, as a meaningful proxy for this purpose (PMID: 22586215). We hope this clarification and restructuring accurately convey the rationale behind our analysis approach.

R2: Line 258: Consider adding the phrase "before pregnancy" after "MNE diet" to emphasize the experimental procedure, in which maternal females were fed the MNE diet at least 12 months prior to pregnancy.

A: Thank you for the suggestion. To better reflect the experimental design, the phrase "before and during pregnancy" was added after "MNE diet" to clarify that maternal females were fed the MNE diet prior to pregnancy, but also throughout pregnancy.

R2: Lines 308, 319, and 351: In the corresponding figure panels, consider adding the appropriate labels below each bar for the CTR and MNE groups, as done in Figures 2 and 3. Additionally, including the measured parameter centered below the CTR and MNE bars could improve clarity and consistency.

A: Thank you for the suggestion. To improve clarity and maintain consistency across figures, we included a legend on the side of the relevant figure panels indicating the appropriate group colors for CTR and MNE. We believe this approach ensures clarity while keeping the visual layout of the figures clean and consistent.

R2: Line 335: If applicable, include the abbreviation in parentheses immediately after "Citrate synthase."

A: The abbreviation "CS" has been added in parentheses immediately after the first mention of "Citrate synthase".

R2: Line 356: In Figure 7C, consider including the term "Mitochondrial DNA" in the axis title, if applicable, to provide a clearer description of the results. Additionally, ensure consistency in axis labeling for relative protein content with respect to the labeling used in Figure 5 for example. Including group labels (CTR and MNE) in the bar plots, where missing, would also improve clarity.

A: Thank you for the helpful suggestions. We have revised the y-axis title in Figure 7A to include the term "Mitochondrial DNA" before “copy number” for improved clarity. Group labels (CTR and MNE) have been added as graphic legends in all figures where multiple parameters are presented. Additionally, the axis labels for relative protein content have been reviewed and standardized across all figures to ensure consistency. While our approach slightly differs from the original suggestion, we are confident it effectively enhances clarity and meets the reviewer’s intent.

R2: Line 373: It would be more informative if the authors discussed the developmental differences at birth between rodents and non-human primates, rather than broadly stating that rodents have low translational value. Rodent models have provided valuable physiological and molecular insights within the DOHaD field, and acknowledging this would offer a more balanced perspective.
Rodents and non-human primates differ markedly in developmental maturity at birth, with primates being precocial, born with more advanced organ development, sensory and motor functions, and a mature brain, while rodents are altricial, requiring significant postnatal growth. Primates typically have longer gestations with fewer offspring per pregnancy (usually singletons or twins), whereas rodents produce larger litters, which impacts resource allocation during development. Additionally, placental structure differs, with primates having a hemochorial placenta more similar to humans, supporting efficient nutrient transfer, while rodents possess a less complex placentation. Metabolically, primates rely more on oxidative metabolism at birth, reflecting greater mitochondrial maturity, whereas rodents undergo a metabolic transition postnatally. These differences in maturity, litter size, placentation, and metabolism influence developmental programming mechanisms and affect the translational applicability of rodent models relative to primates in DOHaD research. (PMID: 27314367, PMID: 17976041, PMID: 24147924, PMID: 16391433)

In this part of the discussion, it would be valuable to cite relevant findings from rodent models that are consistent with intrauterine growth restriction observed in the context of maternal obesity or maternal nutrient excess. Including such references would strengthen the discussion and highlight the translational relevance of these experimental models.

A: We appreciate the comment. In response to feedback from another reviewer, we substantially revised and shortened the first paragraph of the discussion, removing the previous reference to rodent models having low translational value. We fully acknowledge the important contribution of rodent models to the DOHaD field. Following your suggestion, we have also incorporated findings from relevant rodent studies reporting intrauterine growth restriction in the context of maternal obesity or nutrient excess. Specifically, we added a sentence in the third paragraph of the discussion highlighting that, consistent with the placental dysfunction observed in our model, similar outcomes have been reported in rodents. This addition helps to emphasize the translational relevance and complementarity of different animal models in the research area. We consider that these changes do not significantly affect the core content of the discussion.

R2: Lines 405–406: It would be useful to very briefly mention the physiological significance of the ApoB/ApoA1 ratio to help contextualize its relevance within the study’s findings.

A: We have revised the text to briefly mention the physiological significance of the ApoB/ApoA1 ratio, highlighting it as an established marker of atherogenic and, therefore, cardiovascular risk.

R2: Line 407: It would be helpful for readers if the acronym "HAPO" were defined when first mentioned, to ensure clarity and accessibility for a broader audience

A: The full definition of the HAPO study (Hyperglycemia and Adverse Pregnancy Outcome) was added to the text of the manuscript when mentioned.

Reviewer 3 Report

Comments and Suggestions for Authors

This manuscript assesses the effects of maternal overnutrition on late gestation fetal skeletal muscle inflammatory and mitochondrial biogenesis markers in a non-human primate model (baboon). The results demonstrate that inflammatory markers, such as TNFa, are higher, while mitochondrial biogenesis markers like PGC-1a and mitochondrial content are lower in soleus from fetuses exposed to maternal nutrient excess compared with fetuses form control-fed dams. To my knowledge, this has not been previously assessed in baboons. While, the results are novel, the introduction and discussion focus on a broad description of maternal obesity and fetal overgrowth, which is largely different from the weight patterns described in this study. Likewise, similar work on Japanese Macaques is not mentioned. Specific comments below detail these concerns.

There is no discussion of prior work testing mitochondrial respiration and function in skeletal muscle of fetal and juvenile Japanese macaques exposed to maternal obesity (McCurdy).

It is mentioned in the discussion that the rodent models have very low translational value. While non-human primates may have higher translational value than rodents, there are still limitations in the translational value of baboons with respect to tissue develop and intrauterine maturation, among others. Likewise, with respect to dysregulation of skeletal muscle mitochondrial and inflammatory markers, there are strong similarities between rodents and non-human primates (albeit at later ages), and likely humans. This statement should be tempered or supported with evidence.

The patterns of the dam and offspring weight is quite interesting and suggests fetal growth restriction. While mentioned briefly in the discussion, More attention to this and discussion of similarities to other studies throughout is warranted, rather than summarizing the body of human maternal obesity research that largely contains fetal overgrowth.

Were the MDE dams insulin resistant?  Why did they only gain minimal weight in gestation? Was it lower intake of food?

What were the characteristics/treatments of the males? How was conception induced (how was 12 months on maternal diets prior to conception ensured?).

There is a paragraph in the introduction (line 84) that discusses highly detailed information on maternal obesity and differences in offspring insulin signaling cascade. This seems out of place as insulin sensitivity or insulin signaling is not measured in this study.

Typo line 134: “same diet than CTR”

Typo line 416: “cytokines is”

Author Response

REVIEW 3 (R3)

Comments and Suggestions for Authors

R3: This manuscript assesses the effects of maternal overnutrition on late gestation fetal skeletal muscle inflammatory and mitochondrial biogenesis markers in a non-human primate model (baboon). The results demonstrate that inflammatory markers, such as TNFa, are higher, while mitochondrial biogenesis markers like PGC-1a and mitochondrial content are lower in soleus from fetuses exposed to maternal nutrient excess compared with fetuses form control-fed dams. To my knowledge, this has not been previously assessed in baboons. While, the results are novel, the introduction and discussion focus on a broad description of maternal obesity and fetal overgrowth, which is largely different from the weight patterns described in this study. Likewise, similar work on Japanese Macaques is not mentioned. Specific comments below detail these concerns.

A: We thank the reviewer for their thoughtful evaluation and for acknowledging the novelty of our findings. We have carefully revised the relevant sections in response to the reviewer’s valuable suggestions and trust that the changes appropriately address the concerns raised.

R3: There is no discussion of prior work testing mitochondrial respiration and function in skeletal muscle of fetal and juvenile Japanese macaques exposed to maternal obesity (McCurdy).

A: Thank you for pointing out the missing study from McCurdy et al.. Two new paragraphs referencing the study have been added to the discussion section to address their work investigating mitochondrial respiration and function in the skeletal muscle of fetal Japanese macaques exposed to the Western-style diet and maternal obesity. These additions provide context and comparison to some of our findings in the baboon model, especially related to mitochondrial content and biogenesis.

R3: It is mentioned in the discussion that the rodent models have very low translational value. While non-human primates may have higher translational value than rodents, there are still limitations in the translational value of baboons with respect to tissue develop and intrauterine maturation, among others. Likewise, with respect to dysregulation of skeletal muscle mitochondrial and inflammatory markers, there are strong similarities between rodents and non-human primates (albeit at later ages), and likely humans. This statement should be tempered or supported with evidence.

A: Thank you for this important observation. Our intention was to highlight the high translational value of non-human primates. However, we recognize that the original statement was very strictly implying rodent models have low translational value when they have made significant contributions to the DOHaD field and not considering the limitations that exist regarding the translational value of baboons. We acknowledge that baboons are more precocial than humans and that subtle differences in the placental structure may impose limitations not previously considered. In response to feedback from previous reviewers, we have removed this statement from the discussion to avoid any misrepresentation. We appreciate your input, which helped us present a more balanced perspective in the manuscript. We believe that these changes do not substantially alter the overall content of the discussion.

R3: The patterns of the dam and offspring weight is quite interesting and suggests fetal growth restriction. While mentioned briefly in the discussion, More attention to this and discussion of similarities to other studies throughout is warranted, rather than summarizing the body of human maternal obesity research that largely contains fetal overgrowth.

A: Thank you for your comment. We agree that the observed pattern is suggestive of fetal growth restriction or of small for gestational age babies as we mention in our manuscript. To strengthen this point, we have also incorporated examples from animal studies, particularly rodent models, where high-fat maternal diets were associated with reduced placental efficiency and fetal growth restriction. On the other hand, we chose to retain the information on how inadequate GWG can also contribute to reduced fetal weight, as this is often overlooked. There is a general assumption that maternal obesity universally leads to fetal overgrowth or macrosomia. Nevertheless, we have streamlined and summarized some parts of the original text where this is discussed.

R3: Were the MDE dams insulin resistant?  Why did they only gain minimal weight in gestation? Was it lower intake of food?

A: Thank you for your thoughtful questions. We did not directly assess insulin resistance of the animals in this study, so we cannot conclusively determine the insulin sensitivity status of the MNE dams. Moreover, the feeding protocol was detailed in the initial publication describing the model (cited in the manuscript, PMID: 15102068), where food intake was monitored and shown to stabilize early in the protocol. As highlighted in the present manuscript, the diets for both groups were matched for protein content, essential minerals, and vitamins. Given this controlled dietary design and previous findings, we do not believe reduced food intake is a likely explanation for the minimal gestational weight gain observed in the MNE dams.

R3: What were the characteristics/treatments of the males? How was conception induced (how was 12 months on maternal diets prior to conception ensured?).

A: Thank you for your questions. Male baboons used for breeding were maintained on standard chow and were not subjected to any dietary intervention. To ensure group stability during the 12-month nutritional intervention period, groups of up to 16 female baboons were housed with a vasectomized male in outdoor gang cages, allowing for the establishment of a stable social hierarchy and natural physical activity. After the 12-month dietary intervention was completed, the vasectomized male was replaced with a proven fertile male to enable conception. The 12-month exposure to the maternal diet prior to conception was strictly ensured through a well-established individual feeding system, as detailed in the original study describing this model (mentioned in the previous comment). Females were monitored twice daily for overall well-being and three times per week for perineal turgescence and vaginal bleeding to accurately determine the timing of conception. That information was added to the 2.1 subsection of Materials and Methods

R3: There is a paragraph in the introduction (line 84) that discusses highly detailed information on maternal obesity and differences in offspring insulin signaling cascade. This seems out of place as insulin sensitivity or insulin signaling is not measured in this study.

A: Thank you for your helpful comment. We agree that the paragraph was too detailed and have shortened it. However, we decided to keep a brief version because it connects to our previous work in ewes and helps explain the role of inflammation, which is an important part of our study. Even though we did not directly measure insulin signaling, we believe this background supports the context of our findings.

R3: Typo line 134: “same diet than CTR”

A: The typo was corrected. “than” was changed to “as the”.

R3: Typo line 416: “cytokines is”

A: The typo was corrected. “cytokines” was changed to “cytokine”.

Round 2

Reviewer 1 Report

Comments and Suggestions for Authors

As a reviewer, I appreciate the author(s) for addressing the issues highlighted, which helped improve the overall quality of the draft. However, one to two comments still needed to be addressed before moving forward with publication.

Line 33 mentioned the list of genes for expression level, but the symbols are not italicized. Moreover, symbols for PPAR_ are italicized in some sections but not in all; kindly check these issues, although most of them are addressed.

Line 129 mentioned the species name, which I believe should be italicized as well.

Author Response

REVIEW 1 (R1)
Comments and Suggestions for Authors

R1: As a reviewer, I appreciate the author(s) for addressing the issues highlighted, which helped improve the overall quality of the draft. However, one to two comments still needed to be addressed before moving forward with publication:

Answers authors (A): We appreciate the reviewer’s positive feedback and acknowledgment of the improvements made to the manuscript. Below, we provide our replies to the remaining comments to ensure the highest quality of the final version prior to publication.

R1: Line 33 mentioned the list of genes for expression level, but the symbols are not italicized. Moreover, symbols for PPAR_ are italicized in some sections but not in all; kindly check these issues, although most of them are addressed.

A: Thank you for pointing this out. Since we intended to refer to both the protein and transcript levels of PGC1α, we revised the sentence accordingly to read “PGC1α, both protein and transcript levels.”. Additionally, we replaced "PGC1β" with the correct gene symbol PPARGC1B and ensured that all other mentioned gene symbols are properly italicized. Moreover, we have reviewed the formatting of the PPAR symbols throughout the manuscript. In sections where PPARα, PPARδ, and PPARγ are mentioned in the context of their role as transcription factors (i.e., referring to the proteins), we have intentionally kept the symbols in regular font, following standard conventions. In instances where we refer to findings from other studies, we verified the context in the original manuscripts.

R1: Line 129 mentioned the species name, which I believe should be italicized as well.

A: Thank you for your observation. We have revised the text and ensured that the species name (Papio spp.) is correctly italicized in the manuscript, according to taxonomic conventions.

Reviewer 3 Report

Comments and Suggestions for Authors

The authors have addressed my concerns.

Author Response

REVIEW 3 (R3)

Comments and Suggestions for Authors

R3: The authors have addressed my concerns.

A: Thank you for your time and valuable feedback. We appreciate your positive evaluation and are glad our revisions addressed your concerns.